# Demographic and Occupational Determinants of Work-Related Musculoskeletal Disorders: A Cross-Sectional Study

**DOI:** 10.3390/jfmk10020137

**Published:** 2025-04-20

**Authors:** Monika S. Popova, Silviya P. Nikolova, Silviya I. Filkova

**Affiliations:** 1Education Sector “Rehabilitator”, Medical College, Medical University–Varna, 55 Marin Drinov Street, 9000 Varna, Bulgaria; silviq.filkova@mu-varna.bg; 2Department of Social Medicine and Healthcare Organization, Medical University–Varna, 55 Marin Drinov Street, 9000 Varna, Bulgaria

**Keywords:** musculoskeletal diseases, occupational diseases, risk factors, ergonomics, occupational exposure

## Abstract

Work-related musculoskeletal disorders (WMSDs) are a significant public health concern, particularly in professions requiring prolonged static postures. **Objectives:** This study examined the influence of demographic and occupational factors on the WMSD prevalence and provides evidence-based recommendations for workplace health improvements. **Methods**: A cross-sectional study (July–September 2024) surveyed 80 office employees in Varna, Bulgaria, using the Prevent 4 Work (P4Wq) questionnaire. ANOVA and *t*-tests assessed the WMSD prevalence across demographics, while Pearson’s correlations examined associations with age, BMI, and work experience. Data were analyzed in Jamovi v.2.6 (*p* < 0.05). **Results**: The sample (92.5% women, mean age 47.2 years) reported a high WMSD prevalence, with cervical pain (88.8%), lower back pain (83.8%), and shoulder pain (75.0%) being the most common. Work experience, age, and BMI were significantly correlated with WMSD severity, while gender showed no significant associations. **Conclusions**: WMSDs are highly prevalent among office employees, with lower back, neck, and shoulder pain being the most common complaints. Factors such as higher BMI, longer work experience, and increased workload are associated with greater symptom severity. These results emphasize the urgent need for targeted workplace interventions aimed at reducing ergonomic risks, improving posture, and enhancing employee well-being, ultimately fostering a healthier and more productive work environment.

## 1. Introduction

Public health has become a strategic priority at both the European and national levels, with policies increasingly focusing on disease prevention and health promotion [1,2]. Workplace health and safety are central to these efforts, as occupational conditions significantly impact workers’ well-being, productivity, and overall quality of life [3]. Work-related musculoskeletal disorders (WMSDs) have emerged as a major public health concern, necessitating comprehensive prevention and intervention strategies [4,5]. Studies emphasize the importance of understanding how demographic and occupational factors such as age, BMI, and work experience influence the WMSD prevalence [6,7,8,9].

The modern work environment is undergoing rapid transformations due to technological advancements, digitalization, and the automation of both production and administrative processes [10]. The COVID-19 pandemic accelerated these changes, making remote work a dominant employment model [11]. While these developments offer flexibility and efficiency, they also introduce new health risks, including physical inactivity, prolonged screen exposure, and poor posture, all of which contribute to the growing prevalence of WMSDs [11,12,13]. For example, higher BMI has been linked to increased biomechanical stress on joints, while prolonged static postures elevate the risk for spinal and shoulder injuries [7,14,15].

WMSDs refer to a group of conditions affecting muscles, joints, tendons, bones, and nerves, often caused by repetitive movements, sustained improper postures, excessive physical exertion, or direct injuries [16,17]. These disorders are particularly common in occupations that require prolonged sitting, repetitive manual tasks, or work in ergonomically unfavorable conditions [6,18,19]. Symptoms range from mild discomfort to chronic pain and functional impairment, ultimately reducing work capacity and quality of life [20]. Consequently, understanding the demographic and occupational determinants of WMSDs is crucial for designing effective workplace interventions and policy measures.

Globally, WMSDs are one of the leading causes of disability, affecting approximately 1.71 billion people [21]. In Europe, more than half of workers report musculoskeletal complaints, contributing to substantial economic costs—estimated at around 240 billion euros annually [22]. In 2020, WMSDs accounted for approximately 6% of all work-related illnesses, with back pain, shoulder pain, neck pain, and upper limb disorders being the most common manifestations [16,23,24]. Recent meta-analyses have confirmed a high WMSD prevalence in this population, especially in the cervical, lumbar, and shoulder regions [25,26]. Healthcare workers also face significant risks due to sustained physical and psychosocial demands [27,28,29]. Notably, younger workers are increasingly affected, underscoring the need for early prevention strategies [30].

In Bulgaria, musculoskeletal disorders remain a significant public health challenge, contributing to a high number of years lived with disability, particularly among women [31]. Addressing these conditions requires a multidimensional approach, including ergonomic interventions, workplace modifications, and targeted preventive measures [32]. Research consistently demonstrates that investing in WMSD prevention is more cost-effective than treatment, reducing both healthcare expenditures and productivity losses [33,34].

Global health organizations, including the World Health Organization (WHO) and the European Agency for Safety and Health at Work (EU-OSHA), emphasize the need to incorporate the prevention of work-related musculoskeletal disorders (WMSDs) into broader occupational health strategies [21,30]. As the prevalence of WMSDs continues to rise, particularly among workers in professions requiring prolonged static postures, understanding the demographic and occupational factors contributing to this trend has become an area of increasing research interest [6,35]. This study aimed to explore the correlations between WMSDs and key demographic and health-related variables—such as age, body mass index (BMI), and work experience—among office employees. Additionally, it assessed differences in WMSD prevalence across specific demographic and occupational subgroups, providing valuable insights into high-risk profiles and enabling the development of targeted ergonomic and organizational interventions to mitigate injury risks and improve workplace health.

## 2. Materials and Methods

### 2.1. Study Design and Criteria

This observational study employed a cross-sectional design to investigate the associations between demographic and occupational factors and the prevalence of Work-related Musculoskeletal Disorders (WMSDs). This research was conducted from July to September 2024 as part of Project No. 23001 titled “Promoting Healthy Habits and Preventive Measures in the Workplace for Professions Requiring Static Postures”, funded by the Scientific Research Fund at the Medical University—Varna, Bulgaria. This study targeted administrative employees working in professions that require prolonged use of computers, a key factor contributing to the development of WMSDs. This study adhered to the Strengthening the Reporting of Observational Studies in Epidemiology (STROBE) guidelines for reporting observational studies [36].

### 2.2. Setting

Participants were voluntarily recruited though convenience sampling from the workforce of the National Revenue Agency (NRA) in Varna, Bulgaria. Employees were invited to participate in the study through an open call, where they could choose to enroll based on their interest in the study.

### 2.3. Participants

The inclusion criteria for participation were: (1) age between 25 and 65 years, (2) completion of the Work-related Musculoskeletal Disorders Questionnaire (WMSQ), and (3) presence of functional complaints in the upper limbs region, indicating symptoms of musculoskeletal disorders. Exclusion criteria were: (1) history of upper limb injuries, (2) diagnosed neurological or rheumatological conditions, (3) severe systemic diseases such as multiple sclerosis or stage II hypertension, (4) recent surgical intervention on the neck or shoulder, and (5) pregnancy. A total of 500 employees were invited to participate, with 80 agreeing to participate in the study. The sample size was calculated using G*Power (version 3.1), with a power analysis conducted for both an independent samples *t*-test and a one-way ANOVA. For the *t*-test, the analysis was performed with a significance level (alpha) of 0.05, a desired power of 80% (β = 0.80), and an anticipated large effect size (Cohen’s d = 0.58), suggesting a minimum of 74 participants for detecting a statistically significant difference. For the one-way ANOVA, the parameters were a significance level of 0.05, a power of 80%, and an anticipated large effect size (Cohen’s f = 0.50), suggesting a minimum of 74 participants to detect significant group differences. Thus, the sample size of 80 participants was deemed sufficient to achieve the desired power for both statistical analyses. This study was approved by the Research Ethics Committee of the Medical University—Varna (protocol No. 2/04/07/2024), and informed written consent was obtained from all participants prior to data collection, with the study purpose, procedures, and potential risks explained to ensure voluntary participation.

### 2.4. Data Collection Instrument

The data for this study were collected using the *Prevent 4 Work* (P4Wq) questionnaire, a validated self-assessment tool designed to assess the risk of developing Work-related Musculoskeletal Disorders (WMSDs) and their impact on health, work capacity, and quality of life [37]. The questionnaire is divided into nine sections, covering a range of domains: demographic and health data (age, gender, work experience, height, and weight), pain localization (identifying the body regions affected by musculoskeletal pain), physical workload (including task complexity, mental concentration, and time constraints), workplace relationships (measuring interpersonal dynamics), fear and uncertainty related to work, job satisfaction, health status (frequency of physical complaints), psychosocial stress (evaluating anxiety and emotional well-being), kinesiophobia (fear of exacerbating symptoms through physical activity), and ergonomic risks (workplace ergonomics, posture, and repetitive movements). Participants were asked to complete the questionnaire independently, with the guidance of trained research personnel to ensure clarity and accuracy in the responses.

### 2.5. Statistical Analysis

The collected quantitative data were analyzed using Jamovi software (version 2.6.26). Descriptive statistics, including means and standard deviations, were calculated for continuous variables (e.g., age, BMI) to summarize participant characteristics, while frequencies and percentages were computed for categorical variables (e.g., gender, pain location). To examine relationships between continuous variables, Pearson’s correlation coefficients were used. Group differences in WMSD prevalence, based on demographic and occupational factors, were analyzed using one-way analysis of variance (ANOVA), with Welch’s correction applied for unequal variances. An independent samples *t*-test was used to compare the WMSD prevalence between different BMI categories. Post hoc analyses using the Games–Howell test were performed to identify significant group differences. Statistical significance was set at *p* < 0.05. These analyses aimed to identify key risk factors for WMSDs and provide insights into their impact on employee health and well-being.

## 3. Results

### 3.1. Sample Characteristic

The study sample consisted of 80 participants, with a majority of women (92.5%). The participants’ ages ranged from 29 to 61 years, with an average age of 47.2 years (SD = 7.91). On average, participants had 18.2 years of work experience (SD = 8.68), ranging from 1 to 36 years. The mean Body Mass Index (BMI) was 25.3 (SD = 5.71), ranging from 18.4 to 57.3. Regarding the locations of reported symptoms (discomfort, pain, or tension) over the past 12 months, the cervical area was the most commonly affected, reported by 71 participants (88.8%). Lower back pain was the second most frequent symptom, reported by 67 participants (83.8%), followed by shoulder pain, reported by 60 participants (75.0%) (Table 1).

### 3.2. Correlation Analyses

The correlation analysis revealed several significant relationships between the variables assessed in the baseline study on Work-related Musculoskeletal Disorders (WMSDs). A significant positive correlation was observed between work experience and reported WMSD differences (Pearson’s *r* = 0.266, *p* = 0.017), suggesting that individuals with longer work histories reported more pronounced musculoskeletal issues. Similarly, age showed a significant positive correlation with WMSD differences (*r* = 0.223, *p* = 0.046), indicating that older participants tended to report greater musculoskeletal discomfort. Notably, age and work experience were highly correlated (*r* = 0.768, *p* < 0.001), which is consistent with the expectation that older individuals generally have more years of work experience. Body Mass Index (BMI) also demonstrated a significant positive correlation with WMSD differences (*r* = 0.273, *p* = 0.014), indicating that higher BMI levels were associated with increased reports of musculoskeletal discomfort. However, BMI did not significantly correlate with work experience (*p* = 0.175) or age (*p* = 0.127). In contrast, gender showed no significant correlations with WMSD differences or any other variables, with all *p*-values exceeding 0.05 (Table 2).

### 3.3. Group Differences in WMSDs

To explore group differences in Work-related Musculoskeletal Disorders (WMSDs), several statistical analyses were conducted. A one-way ANOVA was performed to examine WMSD differences across three age groups, but no significant variation was found (F(2, 41) = 1.03, *p* = 0.365). The mean WMSD scores were 4.00 (SD = 2.65) for the youngest group, 3.46 (SD = 2.20) for the middle group, and 4.36 (SD = 2.69) for the oldest group, reflecting minimal differences across age categories (Table 3).

An independent samples *t*-test was used to compare WMSD differences between participants with a BMI below 25 and those above 25. No significant differences were found between these groups (t(78) = −0.811, *p* = 0.420). Participants with a BMI below 25 reported a mean WMSD score of 3.83 (SD = 2.24), while those with a BMI above 25 had a slightly higher mean score of 4.29 (SD = 2.92) (Table 3).

In contrast, a one-way ANOVA with Welch’s correction revealed significant differences in WMSD reports based on years of work experience (F(2, 37.3) = 6.96, *p* = 0.003). Participants with less than six years of work experience reported significantly fewer WMSD issues (M = 2.50, SD = 0.93) compared to those with six to twenty years (M = 4.14, SD = 2.42) and those with more than twenty years of experience (M = 4.25, SD = 2.83) (Table 3). Post hoc analyses using the Games–Howell test confirmed these findings, showing significant differences between the <6 years group and both the 6–20 years (*p* = 0.010) and >20 years (*p* = 0.012) groups. However, no significant difference was observed between the 6–20 years and >20 years groups (*p* = 0.982), suggesting that musculoskeletal discomfort plateaus after the initial increase associated with longer work experience.

## 4. Discussion

The present study investigated the associations between demographic and occupational factors and the prevalence of work-related musculoskeletal disorders (WMSDs) among employees engaged in prolonged computer use. Our findings demonstrated significant relationships between work experience, age, and BMI with WMSD prevalence, while no significant association was found for gender. These results have important implications for workplace health interventions and contribute to the growing body of literature on musculoskeletal health in sedentary occupations.

A primary finding of this study was the significant positive correlation between work experience and reported WMSD symptoms, indicating that individuals with longer work histories were more likely to experience musculoskeletal discomfort. This result aligns with previous research suggesting that prolonged exposure to repetitive tasks, poor posture, and static work conditions contribute to the development of musculoskeletal issues over time [38,39]. Similarly, the significant correlation between age and WMSDs supports the notion that cumulative exposure to occupational risk factors increases the likelihood of musculoskeletal discomfort as individuals age [40,41,42,43]. These findings highlight the need for workplace interventions that address ergonomic risk factors, promote posture awareness, and implement weight management programs to reduce the risk of WMSDs among employees with prolonged work experience [5,33].

Additionally, BMI was found to have a significant positive correlation with WMSD symptoms, suggesting that individuals with higher BMI levels reported greater musculoskeletal discomfort. Previous studies have indicated that increased body weight places additional stress on musculoskeletal structures, potentially exacerbating pain and discomfort in the cervical, shoulder, and lower back regions [5,44]. However, BMI did not significantly correlate with work experience or age, suggesting that its impact on WMSDs may be independent of these demographic factors. Given the association between BMI and musculoskeletal discomfort, workplace health initiatives focusing on physical activity and weight management may be beneficial in mitigating WMSD risks [23,45].

Gender, in contrast, was not significantly associated with WMSD prevalence in this study. While some previous research has reported gender differences in musculoskeletal pain, particularly among women due to ergonomic and physiological differences, the lack of significance in this study may be attributed to the small number of male participants [46,47,48]. The predominantly female sample (92.5%) may have limited the ability to detect meaningful gender-based differences.

The high prevalence of WMSDs in the cervical and lower back regions aligns with findings from prior research on sedentary work environments [49], which highlight prolonged sitting, inadequate ergonomic support, and poor posture as key contributors to musculoskeletal discomfort. The ergonomic risks associated with prolonged computer use, such as sustained neck flexion and insufficient lumbar support, likely contributed to the high prevalence of reported symptoms [50]. To mitigate these effects, ergonomic improvements such as adjustable seating, sit–stand desks, and posture training should be prioritized. Encouraging regular breaks and stretching may further reduce strain. Future research should assess the long-term impact of these interventions and explore targeted strategies for high-risk groups [51,52,53].

The study’s findings have important implications for workplace health interventions. Given the significant association between work experience and WMSDs, targeted ergonomic interventions and preventive strategies should be implemented for employees with longer work tenures. Workplace modifications, such as adjustable workstations, ergonomic chairs, and frequent posture breaks, could mitigate the impact of prolonged static postures [5,54]. Additionally, incorporating stretching exercises, physical activity, and educational programs on posture correction may help reduce musculoskeletal complaints among employees [55]. To enhance the effectiveness of these interventions, future research should consider the integration of advanced instrumental evaluation tools, such as tensiomyography (TMG), to assess muscle characteristics in individuals suffering from WMSDs. TMG is a validated, non-invasive diagnostic technique that offers a comprehensive evaluation of muscle function by assessing contractile properties, such as muscle stiffness, endurance, and response times. This approach could yield valuable insights into the pathophysiology of musculoskeletal dysfunction, particularly in the context of low back pain and shoulder injuries [56,57]. By incorporating TMG into workplace health assessments, clinicians and occupational health professionals can gain a deeper understanding of muscle imbalances, endurance ratios, and contractile responses, which are critical for tailoring individualized interventions. The results of a study by So et al. 2019 echo previous research that suggests that workplace exercise programs and ergonomic interventions can improve workers’ occupational health [58].

Despite its contributions, this study has some limitations. First, the cross-sectional design precludes the ability to infer causality, limiting the interpretation of relationships between risk factors and WMSDs. Longitudinal studies would be beneficial in determining the temporal progression of musculoskeletal symptoms in sedentary workers. Second, the use of self-reported data introduces the possibility of recall bias and subjective variation in symptom reporting. Future studies should consider objective measures, such as clinical assessments or biomechanical evaluations, to complement self-reported findings. In future studies, the Nordic Musculoskeletal Questionnaire (NMQ) could be used to investigate the prevalence of musculoskeletal pain in several distinct anatomical regions (neck, upper back, lower back, shoulder, elbow, hand/wrist, hip, knee, and ankle/foot) and its impact on regular work and daily activities in adults with complaints. Lastly, the small sample size and limited male representation may restrict the generalizability of the findings to broader occupational populations.

## 5. Conclusions

This study underscores the importance of demographic and occupational factors in the prevalence of WMSDs among employees engaged in prolonged computer use. The findings emphasize the need for proactive workplace interventions that address ergonomic risks, promote healthy work habits, and mitigate musculoskeletal discomfort among employees. Future research should explore longitudinal trends, incorporate objective assessments, and examine additional workplace variables to develop comprehensive strategies for WMSD prevention and management.

## Figures and Tables

**Table 1 jfmk-10-00137-t001:** Characteristics of the sample.

Category	Value
Gender	
Men	6 (7.5%)
Women	74 (92.5%)
Age	Mean: 47.2 (SD = 7.91), Min = 29, Max = 61
Work Experience (years)	Mean: 18.2 (SD = 8.68), Min = 1, Max = 36
Body Mass Index (BMI)	Mean: 25.3 (SD = 5.71), Min = 18.4, Max = 57.3
Symptoms (discomfort, pain, tension) over the past 12 months	
Cervical area	71 (88.8%)
Shoulder	60 (75.0%)
Lower back	67 (83.8%)
Hip/Thigh	44 (55.0%)
Hand	32 (40.0%)
Wrist/Forearm	28 (35.0%)
Ankle	25 (31.3%)
Elbow	16 (20.0%)

**Table 2 jfmk-10-00137-t002:** Correlations Between Work-related Musculoskeletal Disorders and Demographic/Health Variables.

		WMSDs Difference	Work Experience	Age	BMI
Work experience	Pearson’s r	0.266 *	—		
	*p*-value	0.017	—		
Age	Pearson’s *r*	0.223 *	0.768 ***	—	
	*p*-value	0.046	0 .001	—	
BMI	Pearson’s *r*	0.273 *	0.153	0.172	—
	*p*-value	0.014	0.175	0.127	—
Gender	Pearson’s *r*	0.078	0.093	0.141	0.065
	*p*-value	0.492	0.410	0.213	0.569

Note. * *p* < 0.05, *** *p* < 0.001.

**Table 3 jfmk-10-00137-t003:** Group Comparisons of WMSD Scores Across Age, BMI, and Work Experience.

Variable/Test Used	Groups Compared	Mean (SD)	Statistical Value	*p*-Value
Age/ One-Way ANOVA	Group 1: 29–39 years (N = 17) Group 2: 40–49 years (N = 28) Group 3: 50+ (N = 33)	4.00 (2.65) 3.46 (2.20) 4.36 (2.69)	F(2, 41) = 1.03	0.365
BMI/ Independent Samples *t* test	<25 (N = 46) >25 (N = 34)	3.83 (2.24) 4.29 (2.92)	t(78) = −0.811	0.420
Work Experience/ One-Way ANOVA (Welch’s)	<6 years (N = 8) 6–20 years (N = 36)	2.50 (0.93) 4.14 (2.42)	F(2, 37.3) = 6.96	0.003

## Data Availability

The data presented in this study are available on request from the corresponding author due to privacy and ethical restrictions.

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
