# Peer review of "Demographic and Occupational Determinants of Work-Related Musculoskeletal Disorders: A Cross-Sectional Study"

_jfmk, 2025, doi:10.3390/jfmk10020137_

Round 1
Reviewer 1 Report
Comments and Suggestions for Authors
The aim of the article is to examine the influence of demographic and occupational factors on the prevalence of WMSD in order to propose recommendations for improving health in the workplace. The authors conducted a cross-sectional study of 80 office workers (92.5% female, mean age 47.2) in Bulgaria, using the Prevent 4 Work questionnaire. Classical tests were performed to analyze the prevalence of WMSD according to demographic data, and correlations were used to examine associations with age, BMI and work experience. The results showed a high prevalence in the following body zones: cervical (88.8%), lumbar (83.8%) and scapular (75.0%). Work experience, age and BMI were significantly correlated with WMSD, while gender showed no significant association. The authors conclude that WMSD is widespread among office workers, and is localized to the lower back, neck and shoulder. Factors such as higher BMI, longer work experience and greater workload are associated with greater symptom severity.
The study presented is of interest to the community. The objective is clearly defined and the hypotheses well formulated. The article is well written, the approach is rigorous and the tools used are relevant. However, the following minor comments and recommendations could improve the article:
- In the “introduction” section: A few references on the risks and prevalence of MSDs could be added. Some studies have proposed a meta-analysis of total prevalence and prevalence by body zone, for office workers (doi: 10.1080/10803548.2024.2446107, doi: 10.3233/WOR-230120, doi: 10.1080/10803548.2025.2457186, doi: 10.1093/joccuh/uiae077) and healthcare professionals (doi: 10.1186/s12891-023-06345-6, doi: 10.1177/21650799231185335, doi: 10.1016/j.ijnurstu.2024.104826). This would place the study in a global context. Some of these references could also be used in the discussion to address other aspects of the study, such as high-prevalence body zones, the effect and consequences of the assessment methods used, working conditions, etc.
I also think it's important to include in the introduction section other works that describe and list risk factors for the onset of MSDs on repetitive tasks with posture maintained over a long period (doi: 10.1016/j.heliyon.2024.e25075, doi: 10.3233/WOR-220080, doi: 10.1016/j.jbmt.2024.01.025., doi: 10.3390/healthcare12242560). This will help position the work presented and highlight its originality and contribution to the scientific community.
It might be interesting to add to the discussion some comments on the contribution of the results to the design of ergonomic equipment or to the development of procedures for the prevention of musculoskeletal disorders. The limitations of the study could be extended, in particular by discussing the choice of questionnaire in relation to those generally used, such as the NMQ, and the consequences for the generalization and comparison of results.
Sincerely
Reviewer 2 Report
Comments and Suggestions for Authors
Dear authors,
First of all thank you for the invitation to review your study “ Demographic and Occupational Determinants of Work-Related Musculoskeletal Disorders” I thank the authors for their efforts in producing this study that aligns with my interst. Please find some specific comments below.
TITLE:
-I suggest to add the type of the study
ABSTRACT:
-Please adopt MeSh terms as keywords.
INTRODUCTION
- The introduction is well organize and supported by references, I suggest only to stress the research gap that could justify your work. Moreover, if is possible, I suggest to split your aim in a primary (maybe correlation between work related Muscluloskeletal Disorder and demographic/health varable) and in a secondary (maybe WMSD across age, BMI work experience)
METHOD
- I would add the type of the study
- If the study is a observational study, I would reported all items following the international guidelines (STROBE).
- Please add more details concerning the calculation of sample size
- Please specify the type of Employees, are they administrative?
RESULTS
- I would divide the results into sections following STROBE, this should confer more structure and could be more related to method section.
- I suggest to use only three lines for your tables
DISCUSSION
- I suggest to expand the first paragraph, in order to synthetize all your main findings. Then I suggest a comparison with previous studies.
- Implications for practice and suggestion of future research should be developed more in-dept. For example, it would be interesting to analyse these WMSDs with an instrumental evaluation such as tensiomyograpy that represents a valid tool to assess muscle characteristic in patients with low back pain and other muscle dysfunctions. Please take in to consideration in our discussion this article concerning low back pain (doi: 10.1080/10669817.2023.2252202) and this concerning shoulder (doi: 10.26603/ijspt20201099.)
Round 2
Reviewer 2 Report
Comments and Suggestions for Authors
I thank authors for the work done